# Adjuvant oncolytic virotherapy for personalized anti-cancer vaccination

D. G. Roy [1,2], K. Geoffroy [3,4,5], M. Marguerie[1,2], S. T. Khan[1,2], N. T. Martin[1], J. Kmiecik[6,7], D. Bobbala[6], A. S. Aitken[1,2], C. T. de Souza[1], K. B. Stephenson[6], B. D. Lichty[6,8], R. C. Auer[1,2], D. F. Stojdl[2,6,7], J. C. Bell [1,2] & M.-C. Bourgeois-Daigneault [3,4,5 ✉]

By conferring systemic protection and durable benefits, cancer immunotherapies are emerging as long-term solutions for cancer treatment. One such approach that is currently undergoing clinical testing is a therapeutic anti-cancer vaccine that uses two different viruses expressing the same tumor antigen to prime and boost anti-tumor immunity. By providing the additional advantage of directly killing cancer cells, oncolytic viruses (OVs) constitute ideal platforms for such treatment strategy. However, given that the targeted tumor antigen is encoded into the viral genomes, its production requires robust infection and therefore, the vaccination efficiency partially depends on the unpredictable and highly variable intrinsic sensitivity of each tumor to OV infection. In this study, we demonstrate that anti-cancer vaccination using OVs (Adenovirus (Ad), Maraba virus (MRB), Vesicular stomatitis virus (VSV) and Vaccinia virus (VV)) co-administered with antigenic peptides is as efficient as antigen-engineered OVs and does not depend on viral replication. Our strategy is particularly attractive for personalized anti-cancer vaccines targeting patient-specific mutations. We suggest that the use of OVs as adjuvant platforms for therapeutic anti-cancer vaccination warrants testing for cancer treatment.

[1] Centre for Innovative Cancer Research, Ottawa Hospital Research Institute, Ottawa, ON, Canada. [2] Department of Biochemistry, Microbiology and Immunology, University of Ottawa, Ottawa, ON, Canada. [3] CRCHUM: "Centre Hospitalier de l'Université de Montréal" Research Centre, Montreal, QC, Canada. [4] "Institut du cancer de Montréal", Montreal, QC, Canada. [5] "Département de Microbiologie, Infectiologie et Immunologie, Faculté de Médecine, Université de Montréal", Montreal, QC, Canada. [6] Turnstone Biologics, Ottawa, ON, Canada. [7] Children's Hospital of Eastern Ontario Research Institute, Ottawa, ON, Canada. [8] McMaster Immunology Research Centre, Department of Pathology and Molecular Medicine, McMaster University, Hamilton, ON, Canada. ✉email: marie-claude.bourgeois-daigneault@umontreal.ca

The immune system has evolved to recognize and eliminate pathogens such as viruses. This function has long been exploited for vaccination purposes using viruses modified to encode antigens against which an immune response is desirable. Given their direct oncolytic and immune-stimulating activities, OVs are particularly attractive candidates for such application[1]. One strategy that is currently undergoing clinical testing (NCT02285816 and NCT02879760) is to prime and boost anti-tumor immunity by sequentially administering two different viruses encoding the same tumor antigen[2,3]. Pre-clinical results obtained using Ad and MRB[4] as priming and boosting agents, respectively, demonstrated the possibility of achieving complete responses for a fraction of the animals[3]. While this important study established the potential of the heterologous OV prime-boost vaccine, not all tumors express a shared antigen that is specifically expressed by cancer cells and the vaccination against a single target applies a selection pressure that favors the evolution of unresponsive escape tumor variants[5,6]. Also, the generation of anti-viral immunity is an important limitation that prevents efficient vaccination against a different tumor antigen using the same viruses. Ideally, a single round of treatment would be administered and target all tumor cells despite cancer heterogeneity.

T cells recognizing cancer-specific mutations (so-called mutanome epitopes (Muts)) can be found in patient tumor samples[7] and vaccines targeting these cancer neo-antigens are undergoing clinical testing (NCT02316457, NCT02035956, NCT02149225) after several groups obtained promising pre-clinical and clinical data[8–11]. An OV prime-boost vaccine targeting the tumor mutanome would allow for the direct virus-mediated killing of the cancer cells as well as the generation of anti-tumor immunity; however, the generation of unique viruses tailored to each patient would be time-consuming and thus a considerable limitation to this approach.

In this study, we show that OVs (Ad, MRB, VSV and VV) can be used as adjuvants for anticancer vaccination in prime-boost regimens. By co-administering the viruses with peptides corresponding to Muts, we are able to efficiently immunize animals and confer therapeutic efficacy using murine models of cancer. Our strategy bypasses the need for generating unique viruses for each patient and confers the advantage of being easily adaptable to whichever viral platform is the most desirable for a given indication.

## Results

**OVs can be used as adjuvants for peptide-based vaccination.** In order to determine if OVs can be used as vaccine adjuvants, we first wanted to determine if the priming agent used in the heterologous OV prime-boost vaccine (Ad) could be co-administered with antigens and how its adjuvant activity would compare to that of polyI:C, a commonly-used adjuvant[8]. Following the treatment schedule depicted in Fig. 1a, we observed that Ad-DCT (virus engineered to encode the full dopachrome tautomerase (DCT) protein), Ad+DCT (virus co-administered with DCT peptide) and polyI:C + DCT (polyI:C co-administered with DCT peptide) all induced comparable immune responses (Fig. 1b), therefore demonstrating that Ad can act as an adjuvant for anticancer vaccination. We next sought to determine if the boosting virus used in the heterologous OV prime-boost vaccine (MRB) also had adjuvant activity. To do so, we vaccinated tumor-bearing animals with MRB-antigen (full protein) vs MRB + antigen (peptide) using either DCT or Ova as model antigens. Interestingly, our results show that while both MRB-DCT and MRB-Ova failed at inducing antigen-specific immunity, which is in line with a previous report[3], MRB could efficiently trigger anti-

DCT and -Ova immune responses when used as a vaccination adjuvant (Fig. 1c, left and right panels, respectively). Notably, we obtained similar results using VSV and VV (Fig. 1d). In order to determine if pre-mixing the virus with peptide affected its location upon injection, we performed a biodistribution analysis of MRB-Ova with or without the co-administration of Ova peptide (Fig. 1e). Our results show that 24 h post-injection, the virus was found in comparable amounts in the lungs, spleens, livers and brains of the animals from both treatment groups. To determine if the peptides were binding to MRB, we performed a fractionation experiment in which a MRB + peptide mixture was centrifuged through a 50 kDa filter. To facilitate detection, we used peptides corresponding to the sequence of the myc tag. Given that the myc peptide is 10 amino acids long, its size is smaller than 50 kDa while MRB virions are bigger than 50 kDa. Our results show that while MRB was retained by the filter, we did not detect any peptide in the retentate along with MRB, and that the peptide was exclusively found in the filtrate, which indicates that it does not bind to the virus (Supplementary Fig. 1).

A previous study by Bridle et al. investigated the mechanism of action of a heterologous oncolytic virus prime-boost vaccine using Ad-antigen and VSV-antigen administered following the same treatment regimen we are using[12]. The study revealed that the infection of splenic B cells by the boosting vector (VSV) was important for vaccination efficacy. We therefore tested if MRB mixed with peptide could also infect splenic B cells. To facilitate detection of infected cells, we used a GFP-encoding variant of MRB alone or pre-mixed with Ova peptides. Mice were injected IV and spleens were collected 1.5 h later. To allow enough time for GFP expression, the cells were then cultured ex-vivo for an additional 4.5 h prior to staining and analysis. We found a small percentage of splenic B cells (CD19+, B220+) to be GFP+ in both treatment groups (Fig. 1f and Supplementary Fig. 2a), therefore indicating that MRB can also infect splenic B cells and that the presence of peptides did not impair this ability. Further characterization of the virus-infected B cells based on the expression of the markers CD21/35 and CD23 showed that both marginal zone (MZ), follicular, as well as other (non-MZ and non-follicular) B cells were infected (Fig. 1g and Supplementary Fig. 2b), with most of the GFP+ B cells being of the follicular type and no difference being observed between the two groups. This result is consistent with the results obtained using VSV-antigen as a boosting agent in a previous study[12]. Taken together, our data demonstrate that several OVs are efficient vaccination adjuvants and that the addition of free peptides to MRB mixtures does not modify the biodistribution of the virus or its ability to reach and infect discrete subsets of splenic B cells.

We then wanted to determine if OVs could be used as vaccination adjuvants for immune boosting in the heterologous virus prime-boost setting. To do so, we primed tumor-bearing animals with Ad-DCT and boosted the immune response 7 days later using VV, VSV or MRB together with DCT peptide (see treatment schedule, Fig. 2a). Our results show that VV, VSV and MRB can all efficiently boost antigen-specific immunity (Fig. 2b). Interestingly, further experiments showed that the irradiation of MRB (which inactivates the virus) did not affect the magnitude of the immune response generated, thus demonstrating that viral replication is not required for adjuvanticity (Supplementary Fig. 3). When testing different routes of administration, we found that MRB + DCT could efficiently boost the immune response when administered intravenously (IV), intratumorally (IT) as well as intramuscularly (IM) (Supplementary Fig. 4). Importantly, MRB could not be used both as a priming and boosting adjuvant platform as the immune responses mounted in these conditions were comparable to the one obtained following MRB + DCT priming only (Supplementary Fig. 5). We next compared the

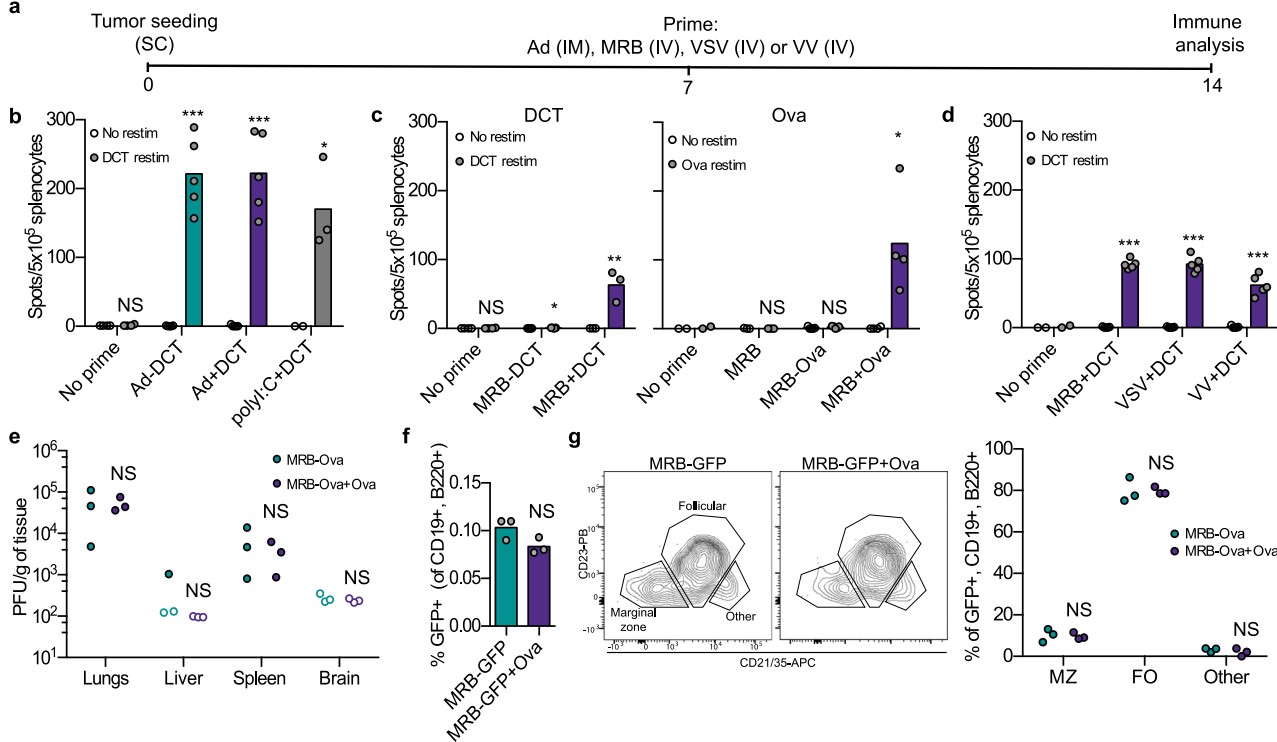

**Fig. 1 OVs are effective as adjuvants for immunization. a** Treatment schedule used in (**b–d**). All mice were bearing SC B16F10-Ova tumors. **b** IFNγ ELISPOT analysis of splenocytes from mice immunized with Ad-DCT or Ad or polyI:C co-administered with DCT peptide (all IM) (from left to right; n = 4, 5, 5, and 3); **c** MRB-DCT, MRB co-administered with DCT peptide (left panel) or MRB-Ova or MRB co-administered with Ova peptide (right panel) (all IV) (from left to right; n = 4, 5, 3, 3, 3, 5 and 4) or; **d** MRB, VSV or VV co-administered with DCT peptide (all IV) (from left to right; n = 2, 5, 5 and 5). For **b–d**, the statistical analyses refer to the comparison between the corresponding ex-vivo "No restim" and "restim" conditions. NS: p > 0.05, *: p < 0.05, **: p < 0.01, ***: p < 0.001 (unpaired two-tailed t-test). **e** Biodistribution analysis of C57BL/6 mice injected IV with MRB-Ova with or without co-administration of Ova peptide for 24 h (n = 3). **f** Flow cytometry analysis of GFP+ splenocytes from mice injected with MRB-GFP with or without Ova peptide. The spleens were collected 1.5 h post-injection, dissociated and cultured ex-vivo for 4.5 h prior to staining and analysis. The graph shows the percentage of live B cells (CD19+, B220+) that are GFP+ (n = 3). **g** Flow cytometry analysis of GFP+, B220+, CD19+ cells from (**f**) (n = 3). The contour plots show the gating strategy to distinguish different subsets of B cells and the right graph shows the quantification of the marginal zone (MZ), follicular (FO) and other B cell populations. NS: p > 0.05 (unpaired two-tailed t-test). Source data are provided as a Source Data file. Exact p values can be found in the Source Data.

adjuvant ability of MRB to that of vaccine adjuvants used in patients. We compared both Alhydrogel (aluminium hydroxide (alum)-based) and Addavax (oil-in-water emulsion similar to MF59)[13] to MRB as boosting adjuvants in our heterologous vaccination regimen. We found MRB to be the best adjuvant to boost CD8 T cell immunity 7, 21, and 28 days post-boost (Supplementary Fig. 6). A modest, but insignificant boosting effect was observed for Addavax, while Alhydrogel-boosted animals showed no improvement compared to the group that received only the prime. Taken together, our data show that MRB is an efficient adjuvant for immune boosting in the heterologous oncolytic virus prime-boost setting.

We then compared MRB-antigen to MRB + antigen vaccinations upon Ad-antigen priming and found that equivalent antigen-specific immune responses were mounted using both boosting strategies (Fig. 2c and d). Flow cytometry analysis revealed that 15–20% of splenic CD8 T cells were antigen-specific using both prime-boost approaches (Fig. 2e), further confirming that MRB co-administered with an antigenic peptide is as effective at boosting anti-tumor immunity as MRB encoding the same antigen. Importantly, we obtained similar results using the clinical trial candidate viruses Ad-E6/E7 and MRB-E6/E7 (Fig. 2f), thus confirming the effectiveness of MRB as an adjuvant for immune boosting.

We next wanted to characterize the CD8 T cells induced by boosting with adjuvant MRB vaccination. For these experiments, we used tumor-free animals to allow for sample collection at later time points at which tumor-bearing animals from control groups would be dead. All the mice were primed with Ad-Ova on day 0 and boosted or not with MRB co-administered with Ova on day 7 (see treatment schedule in Fig. 3a). Impressively, more than 60% of blood CD8 T cells were Ova-tetramer+ 7 days post-boost compared to 20% for animals that received the prime only (Fig. 3b, top graph). Given that more CD8 T cells were detected after boosting, the difference between the two groups was further amplified when looking at the percentage of live blood cells that were Ova-specific CD8 T cells: a difference of nearly 20-fold between the 2 groups (from 2 to 38% with MRB boosting) (Fig. 3b, bottom panel). Similar analyses performed 3 and 4 weeks after boosting (days 28 and 35, respectively) revealed that the percentage of CD8 T cells that were Ova-specific remained at similar levels over time (Fig. 3b, top graph) while the percentage of total blood cells that were Ova-specific CD8 T cells declined over time, but still persisted to significantly higher levels compared to controls (Fig. 3b, bottom panel). The splenocytes were also analyzed for cytokine production upon ex-vivo peptide re-stimulation at day 35. As expected, we found the MRB-boosted animals to have improved capacities at producing both IFNγ and TNFα when stimulated with Ova peptides (Fig. 3c and d). With the objective of further characterizing the antigen-specific CD8 T cells induced by our vaccine, we first assessed the expression of CD127 and KLRG1, which are surface markers that allow for the

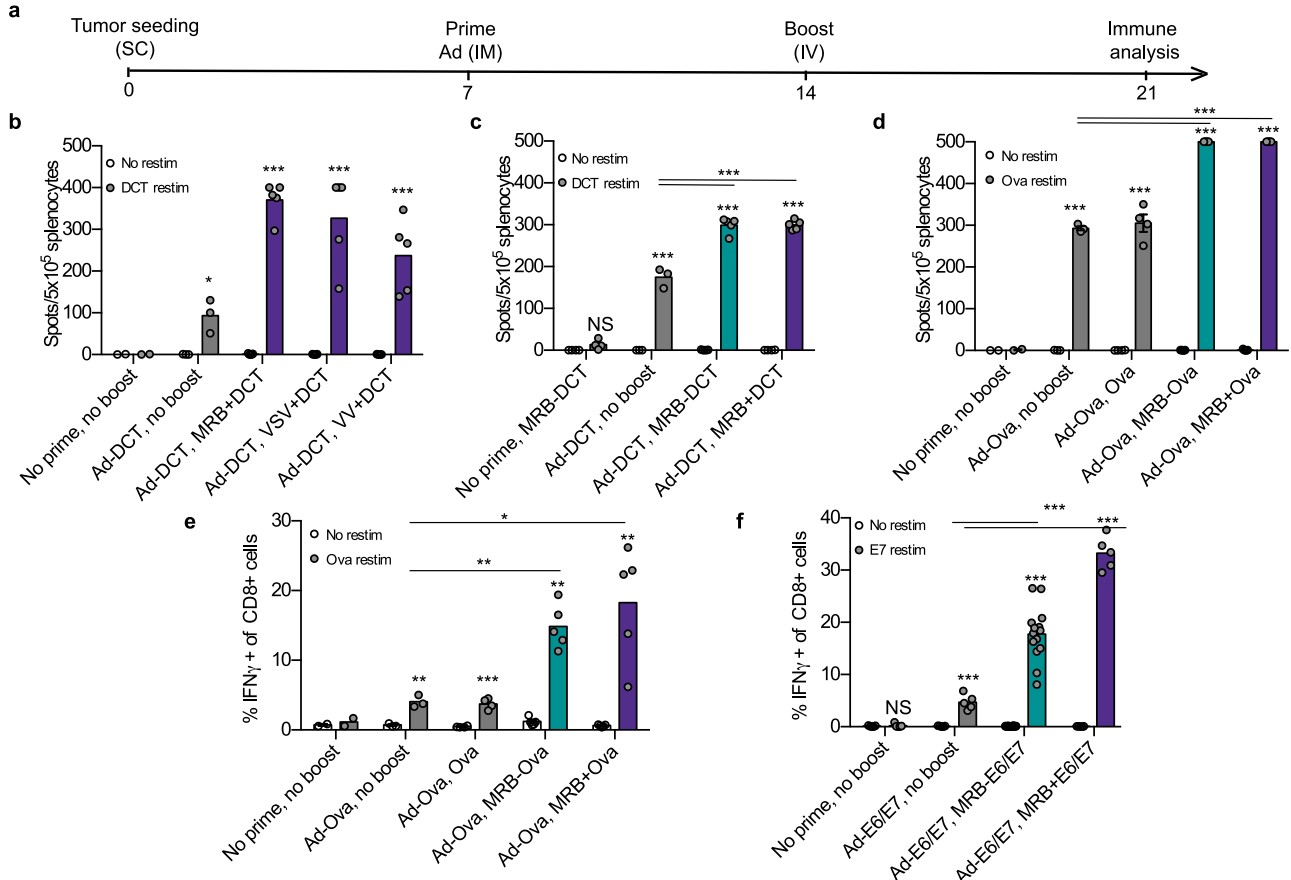

**Fig. 2 OVs are effective adjuvants for immune boosting. a** Treatment schedule used in this study. **b** IFNγ ELISPOT analysis of splenocytes from mice primed with Ad-DCT and boosted with either VV, VSV or MRB co-administered with DCT peptide (from left to right; n = 2, 3, 5, 5 and 5); **c** primed with Ad-DCT and boosted with MRB-DCT or MRB co-administered with DCT peptide (MRB + DCT) (from left to right; n = 4, 4, 5 and 5); **d** primed with Ad-Ova and boosted with MRB-Ova, MRB co-administered with Ova peptide (MRB + Ova) or Ova peptide only (from left to right; n = 2, 3, 4, 5 and 5). **e** Flow cytometry analysis of splenocytes from the same experiment as in (**d**) (from left to right; n = 2, 3, 4, 5 and 5) or (**f**) from mice primed with Ad-E6/E7 and boosted with MRB-E6/E7 or MRB co-administered with E6/E7 peptides (from left to right; n = 5, 5, 14 and 5). Unless indicated otherwise, the statistical analyses refer to the comparison between the corresponding ex-vivo "No restim" and "restim" conditions. NS: p > 0.05, *: p < 0.05, **: p < 0.01, ***: p < 0.001 (unpaired two-tailed t-test). Source data are provided as a Source Data file. Exact p values can be found in the Source Data.

discrimination between memory precursor effector cells (MPECs: KLRG1$^{lo}$, CD127$^{hi}$), double-positive effector cells (DPECs: KLRG1$^{hi}$, CD127$^{hi}$), short-lived effector cells (SLECs: KLRG1$^{hi}$, CD127$^{lo}$) and early effector cells (EECs: KLRG1$^{lo}$, CD127$^{lo}$). We found the different types of Ova-specific CD8 T cells to be of similar proportions with and without MRB boosting, with the biggest fraction belonging to the SLECs sub-population and remaining stable over time (Fig. 3e and f). Although the MRB-boosted mice had slightly more DPECs and fewer SLECs, the differences observed between the treatment groups were not statistically significant. When looking at the absolute number of cells in the spleen, we found more of all sub-populations with MRB + antigen immune boosting (Fig. 3g), a finding that is consistent with the increased number of Ova-specific T cells detected for the boosted mice. We further characterized the cells by assessing the expression of CD62L and CD44, which allow for the identification of naïve: CD44$^-$, CD62L$^+$, central memory (CM: CD44$^+$, CD62L$^+$), effector memory (EM: CD44$^+$, CD62L$^-$) and double negative (DN: CD44$^-$, CD62L$^-$) cells (Fig. 3h, left panel). We found that most of the Ova-specific CD8 T cells from both treatment groups were EM cells (Fig. 3h (right panels), I (top panel) and Supplementary Fig. 7). Furthermore, the phenotype of the cells remained stable over time and the absolute numbers, while higher for all but naïve cells, were only significantly

increased with adjuvant MRB boosting for the EM subtype (Fig. 3i, bottom panel). Taken together, our results show that MRB co-administered with antigen efficiently boosts antigen-specific immunity upon Ad-antigen immune priming and generates important pools of memory cells, mostly EMs and CMs.

**OVs can be used as adjuvants for heterologous virus prime-boost vaccines.** Given that prime-boost vaccination regimens induce greater immune responses compared to single immunizations, we next tested whether both Ad and MRB could be co-administered with peptides and how the vaccination efficiency of this prime-boost strategy compares to the one using the viruses encoding the same antigens (treatment schedule shown in Fig. 4a). Using either DCT or ovalbumin (Ova) as antigens, our results clearly show that OVs + peptides are as effective as OVs encoding the antigens at inducing antigen-specific immunity in this context (Fig. 4b, left and right graphs, respectively). We then tested if our strategy could confer therapeutic benefits to animals bearing established B16F10 lung tumors and interestingly found that 30% of the animals from the OVs + antigen prime-boost group were cured by the treatment (Fig. 4c), while the two viruses administered without antigens failed at providing therapeutic benefits in this aggressive tumor model. When comparing the

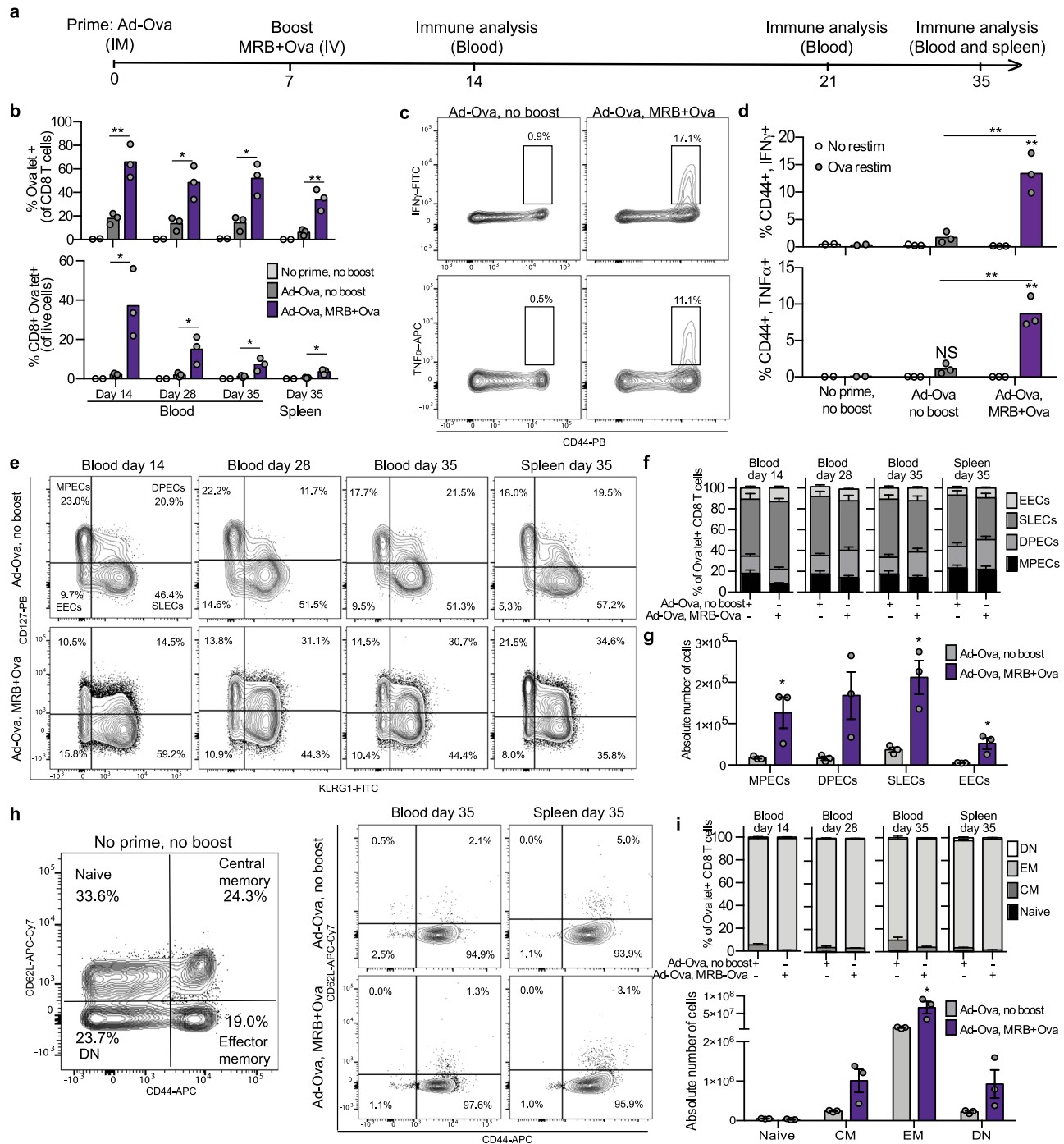

adjuvant OV prime-boost with the vaccine using antigen-encoding viruses, both vaccines provided equivalent therapeutic efficacies (Fig. 4d), a finding that is in line with the induction of equivalent anti-tumor immune responses by the two vaccines (Fig. 4b). Given that we found antigen-specific CD8+ T cells to be induced upon vaccination (Fig. 2e, f and 3), we next sought to determine if depleting CD8 T cells would abolish the therapeutic efficacy of the adjuvant OV vaccine. As expected, the protection conferred by Ad + Ova and MRB + Ova vaccination was completely lost with CD8 depletion (Fig. 4e). Taken together, these results show that OVs can be used as adjuvants for anticancer vaccination and improve survival of tumor-bearing animals in a CD8-dependent manner.

**OVs can be used as personalized cancer vaccine platforms**. In order to determine if OVs could be suitable adjuvants for vaccines targeting tumor neo-epitopes (Muts), we first measured the immune response generated upon vaccination against previously-described Muts[8,14] from the B16F10 cell line using polyI:C as an adjuvant (treatment schedule shown in Fig. 5a). As done in a previous study[8], we used synthetic 27-mer peptides with the different mutations located at position 14 in order to allow for the processing and presentation of all possible epitope variants of the mutations. As expected, different Muts induced antigen-specific immune responses that were superior against the mutations compared to the wild-type versions of the same peptides (Fig. 5b). Next, we sought to determine if MRB and polyI:C could

**Fig. 3 Adjuvant OV boost vaccination generates memory CD8 T cells. a** Treatment schedule used in this study. **b** Flow cytometry analysis of splenocytes and blood cells from tumor-free mice primed with Ad-Ova and boosted or not with MRB co-administered with Ova peptide 7 days later. Blood samples were collected on days 14, 28 and 35 and splenocytes were also analyzed on day 35. The bar charts show the percentage of Ova-tetramer+ cells (within the CD8+ live cell population) (top panel) or of live cells that are CD8+, Ova-tetramer+ (bottom panel) ($n = 2$ (no prime, no boost) and 3 (Ad-Ova, no boost and Ad-Ova, MRB + Ova)). **c** Contour plots showing CD44 expression and IFNγ or TNFα production by CD8+ splenocytes (day 35) re-stimulated ex-vivo with Ova peptides. **d** Quantification of the percentage of CD8+ cells that are CD44+, IFNγ+ or CD44+, TNFα+ with and without ex-vivo re-stimulation $n = 2$ (no prime, no boost) and 3 (Ad-Ova, no boost and Ad-Ova, MRB + Ova). **e** Contour plots showing the expression of KLRG1 and CD127 by Ova-tetramer+ CD8 T cells. The graphs also delineate different T cell subpopulations: memory precursor effector cells (MPECs), double-positive effector cells (DPECs), short-lived effector cells (SLECs) and early effector cells (EECs) ($n = 2$ (no prime, no boost) and 3 (Ad-Ova, no boost and Ad-Ova, MRB + Ova)). **f** Quantification of the populations identified in (**e**) for all mice ($n = 3$). Data are presented as mean values ± SEM. **g** Absolute number of MPECs, DPECs, SLECs and EECs from the spleen of the mice at day 35 ($n = 3$). Data are presented as mean values ± SEM. **h** Representative contour plots showing different T cell populations based on CD44 and CD62L expression of splenocytes from untreated animals (left panel) or of CD8+, Ova-tetramer+ cells from the blood and the spleen of both treatment groups at day 35 (right panels). The graphs also show the gating strategy used to distinguish between naïve, central memory (CM), effector memory (EM) and double negative (DN) CD8+ T cells ($n = 2$ (no prime, no boost) and 3 (Ad-Ova, no boost and Ad-Ova, MRB + Ova)). **i** Quantification of the populations identified in (**g**) for all mice at different time points (top panel) and absolute number of naive, CM, EM and DN CD8+, Ova-tetramer+ cells from the spleen of mice at day 35 (bottom panel) ($n = 3$). Data are presented as mean values ± SEM. Unless indicated otherwise, the statistical analyses refer to the comparison between the corresponding ex-vivo "No restim" and "restim" conditions. NS: $p > 0.05$, *: $p < 0.05$, **: $p < 0.01$, ***: $p < 0.001$ (unpaired two-tailed $t$-test). Source data are provided as a Source Data file. Exact $p$ values can be found in the Source Data.

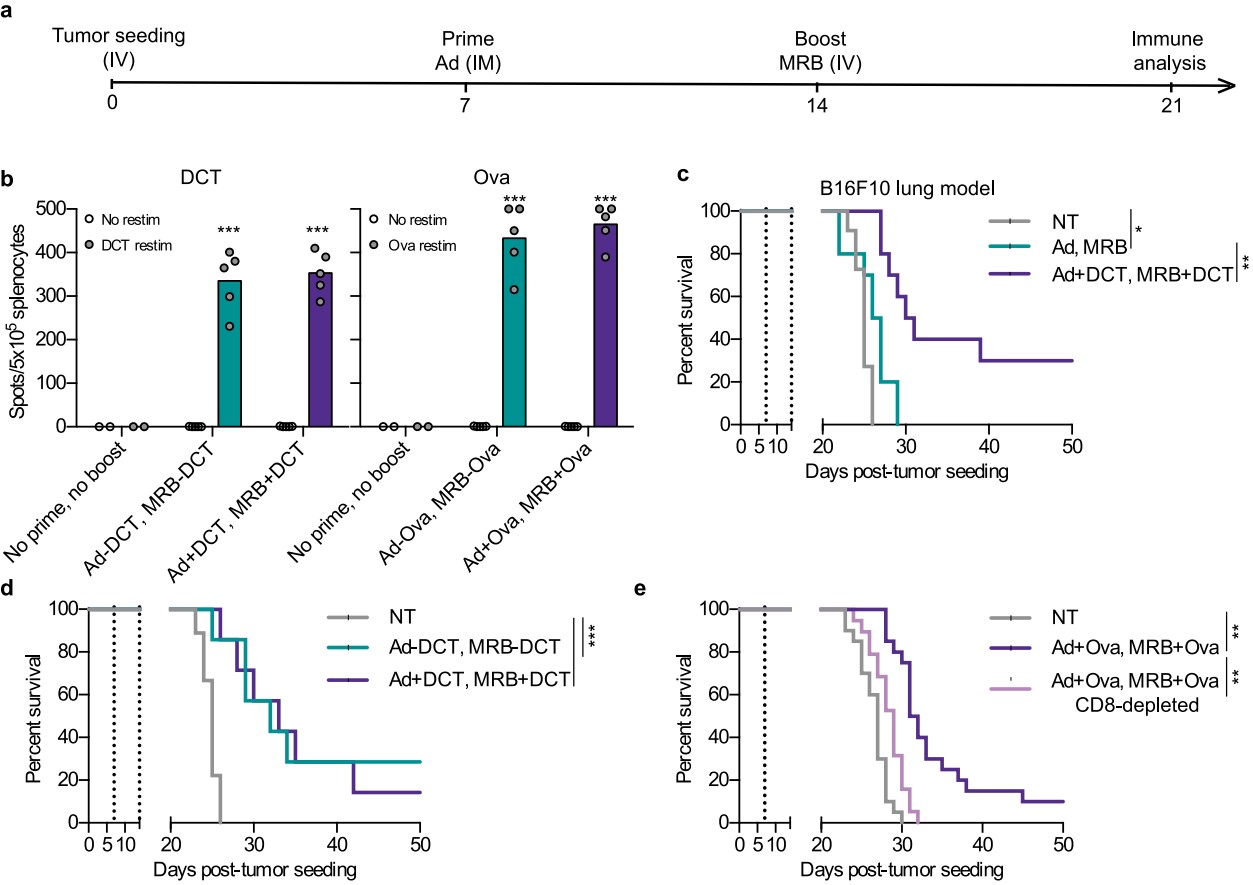

**Fig. 4 OVs can function as both priming and boosting adjuvants in heterologous vaccination regimens. a** Treatment schedule used in this study. **b** IFNγ ELISPOT analysis of splenocytes from mice primed with Ad-DCT on day 7 and boosted with MRB-DCT on day 14 or primed and boosted with Ad and MRB co-administered with DCT peptide (left graph); or primed with Ad-Ova on day 7 and boosted with MRB-Ova on day 14 or primed and boosted with Ad and MRB co-administered with Ova peptide (right graph) (from left to right; $n = 2$, 5 and 5). The statistical analyses refer to the comparison between the corresponding ex-vivo "No restim" and "restim" conditions. NS: $p > 0.05$, ***: $p < 0.001$ (unpaired two-tailed $t$-test). Kaplan–Meier survival analyses of B16F10 lung tumor-bearing mice treated with; **c** Ad (day 7) and MRB (day 14) or Ad and MRB co-administered with DCT peptide ($n = 10$ (Ad, MRB and Ad+DCT, MRB + DCT) and 11 (NT)); **d** Ad-DCT on day 7 and MRB-DCT or MRB co-administered with DCT peptide on day 14 or ($n = 7$ (Ad-DCT, MRB-DCT and Ad+DCT, MRB + DCT) and 9 (NT)); **e** Ad-Ova on day 7 and MRB co-administered with Ova peptide on day 14 with or without CD8 depletion ($n = 20$ (NT and Ad+Ova, MRB + Ova) and 19 (Ad + Ova, MRB + Ova -CD8), 2 experiments combined). $p > 0.05$, *: $p < 0.05$, **: $p < 0.01$, ***: $p < 0.001$ (Mantel–Cox test, two-sided). Source data are provided as a Source Data file. Exact $p$ values can be found in the Source Data.

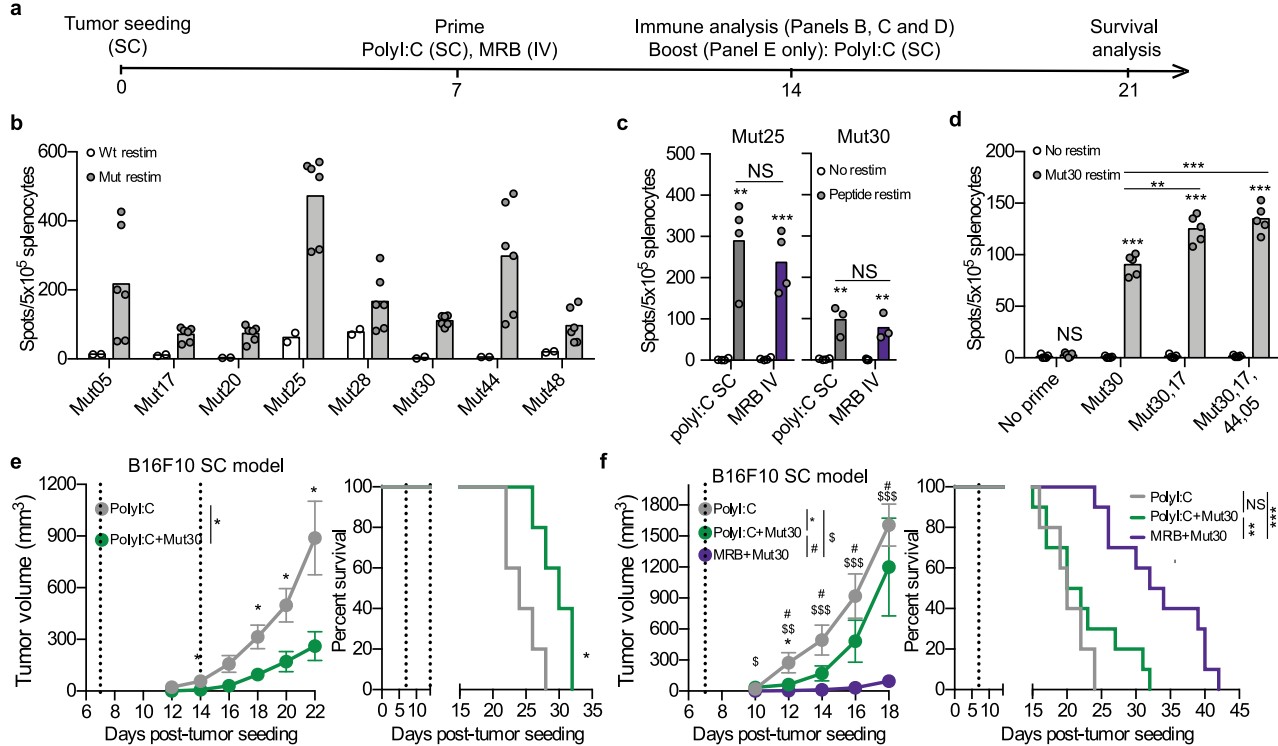

**Fig. 5 OVs can be used as adjuvants for vaccination against cancer neo-epitopes. a** Treatment schedule used in this study. IFNγ ELISPOT analyses of splenocytes from mice immunized with; **b** polyI:C co-administered with different B16F10Muts (IM) and re-stimulated ex-vivo with the same or the corresponding non-mutated peptides ($n = 2$ (WT restim) and 6 (Mut restim)); **c** polyI:C or MRB co-administered with different B16F10Muts and re-stimulated ex-vivo with the same peptides or (from left to right; $n = 4, 4, 3$ and 3); **d** polyI:C co-administered with 1, 2 or 4 different Muts (100 μg each) and re-stimulated ex-vivo with Mut30 ($n = 5$). Unless indicated otherwise, the statistical analyses refer to the comparison between the corresponding "No restim" and "restim" conditions. NS: $p > 0.05$, *: $p < 0.05$, **: $p < 0.01$, ***: $p < 0.001$ (unpaired two-tailed $t$-test). Tumor growth (left panels) and Kaplan–Meier survival analyses (right panels) of B16F10 SC tumor-bearing mice immunized with; **e** polyI:C ± Mut30 on days 7 and 14 or ($n = 5$), Data are presented as mean values ± SEM.; **f** polyI:C or MRB + Mut30 on day 7 ($n = 10$). Data are presented as mean values ± SEM. Tumor growth analyses: NS: $p > 0.05$, *: $p < 0.05$, ***: $p < 0.001$ (unpaired multiple two-tailed $t$-tests) and survival analyses: NS: $p > 0.05$, *: $p < 0.05$, **: $p < 0.01$, ***: $p < 0.001$ (Mantel–Cox test, two-sided). Source data are provided as a Source Data file. Exact $p$ values can be found in the Source Data.

both induce similar anti-Mut immune responses. First, we determined the route of administration of polyI:C that conferred the best vaccination efficacy. Our results show that the optimal injection routes for polyI:C + peptides were IM and sub-cutaneously (SC) (Supplementary Fig. 8a). Given that MRB is usually administered IV in our vaccination model, we compared the adjuvant activity of polyI:C and MRB co-administered with peptides SC or IV. When comparing these vaccines head-to-head, we observed that the antigen-specific immune responses were equivalent for both adjuvants when immunizing against DCT (Supplementary Fig. 8b), therefore demonstrating once again that the adjuvant activity of MRB compares to that of polyI:C. Furthermore, we obtained similar findings when immunizing against two different Muts (Fig. 5c). Given that our ultimate objective is to immunize against a cocktail of antigens, we next tested whether the vaccination against 1, 2 or 4 different Muts would compro-mise the vaccination efficacy against individual peptides. Importantly, our results showed that the antigen-specific immune response was unaffected by the number of different peptides used for vaccination (Fig. 5d).

We then wanted to measure the therapeutic benefits provided by adjuvant MRB vs polyI:C vaccination against Mut30 in the B16F10 SC tumor-bearing mice. First, we confirmed that, as previously published by another group[8], the anti-Mut30 ther-apeutic vaccination could control tumor growth and slightly extend survival (Fig. 5e, left and right panels, respectively). Next, we compared the therapeutic efficacy of polyI:C and MRB

adjuvant vaccinations and found that a single dose of the MRB vaccine was sufficient to control tumor growth and extend the median survival of the animals by 10 days compared to polyI:C-based vaccination (Fig. 5f, left and right panels, respectively). Taken together, our data demonstrate that MRB and polyI:C have equivalent vaccine adjuvant activities against tumor neo-epitopes, but that the therapeutic benefits are superior when using an OV for anticancer vaccination.

Given that our objective is to prevent vaccination-induced cancer immuno-editing by immunizing against multiple antigens, we first wanted to identify multiple targetable cancer neo-antigens. In order to select which Muts to incorporate in our B16F10-tailored vaccine, we tested the therapeutic potential of previously-described neo-epitopes[14] using polyI:C as an adjuvant to prime and boost anti-tumor immunity (treatment schedule shown in Fig. 6a). We found that 4 of the neo-epitopes tested (B16Mut20, B16Mut30, B16Mut44, and B16Mut48) could successfully slow tumor growth in this therapeutic setting (Fig. 6b). We then tested whether vaccines consisting of OVs +Muts could confer therapeutic benefits to animals bearing established tumors. To do so, we pre-mixed Ad or MRB with the 4 protective neo-epitopes and vaccinated mice bearing established B16F10 lung tumors. An ELISPOT analysis revealed that the vaccinated animals mounted immune responses against all 4 Muts (Fig. 6c). Furthermore, and as observed in Fig. 4c, the mice treated with Ad and MRB alone were not protected against the B16F10 lung tumors (Fig. 6d). On the other hand, the

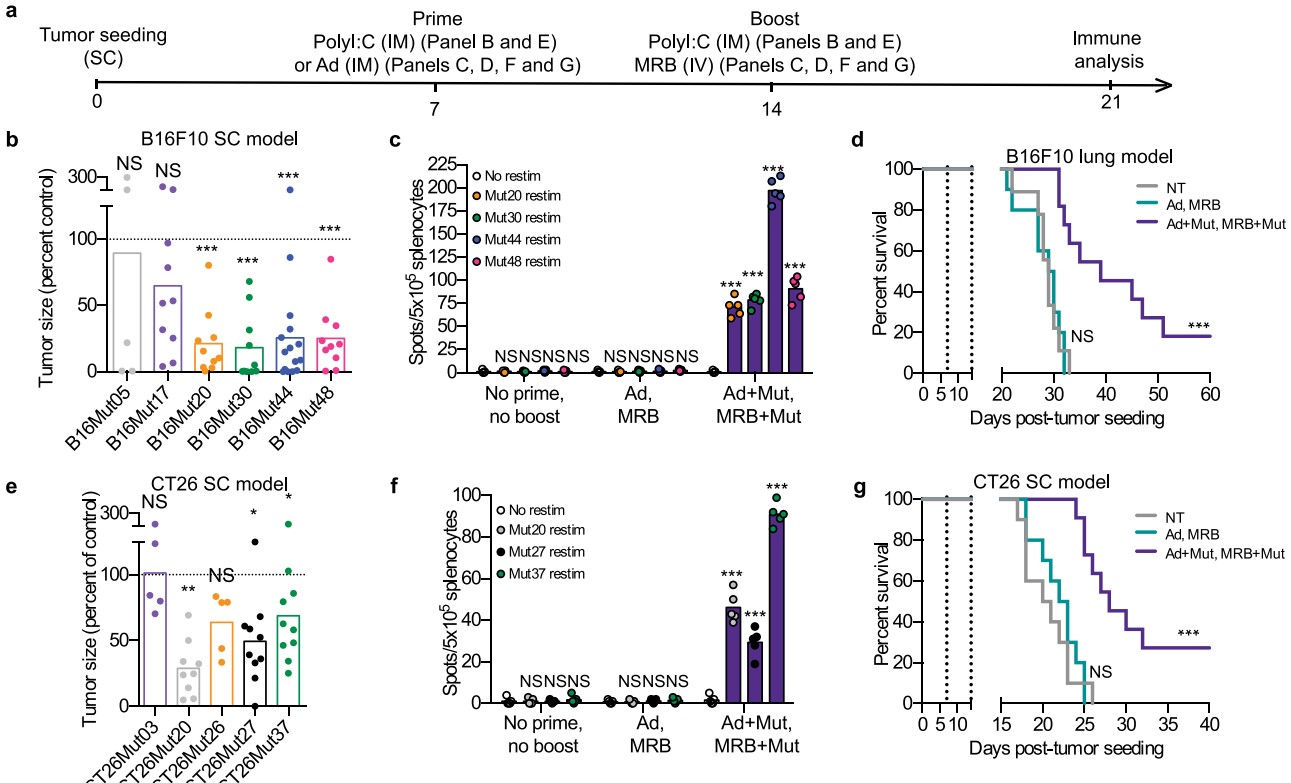

**Fig. 6 Adjuvant OV vaccination against Muts improves outcome. a** Treatment schedule used in this study. **b** Mice bearing established SC B16F10 tumors were treated IM with polyI:C and the indicated peptide on days 7 and 14 post-tumor seeding. All tumors were measured on day 21. The tumor volumes are expressed as relative to the average tumor volume of control mice (treated with polyI:C only) (from left to right; $n = 5, 10, 10, 4, 10, 15$ and $10$). NS: $p > 0.05$, ***: $p < 0.001$ (unpaired two-tailed $t$-test). **c** IFNγ ELISPOT ($n = 5$) and (**d**) Kaplan–Meier survival analysis ($n = 10$) of B16F10 lung tumor-bearing mice treated with Ad and MRB or Ad and MRB co-administered with B16Mut20, B16Mut30, B16Mut44 and B16Mut48. ELISPOT: NS: $p > 0.05$, ***: $p < 0.001$ (unpaired two-tailed $t$-test) and survival: NS: $p > 0.05$, ***: $p < 0.001$ (Mantel–Cox test, two-sided). **e** Mice bearing established SC CT26 tumors were treated and analyzed as in (**b**) (from left to right; $n = 9, 5, 5, 10$ and $10$). NS: $p > 0.05$, *: $p < 0.05$ (unpaired two-tailed $t$-test). **f** IFNγ ELISPOT ($n = 5$) and (**g**) Kaplan–Meier survival analysis ($n = 10$) of SC CT26 tumor-bearing mice treated on days 7 and 14 post-tumor seeding with Ad and MRB or Ad and MRB co-administered with CT26Mut20, CT26Mut27 and CT26Mut37. ELISPOT: NS: $p > 0.05$, ***: $p < 0.001$ (unpaired two-tailed $t$-test) and survival: NS: $p > 0.05$, ***: $p < 0.001$ (Mantel–Cox test, two-sided). Source data are provided as a Source Data file. Exact $p$ values can be found in the Source Data.

heterologous OV + Muts prime-boost vaccine could cure 20% of the animals and extend survival of the rest of the cohort. In order to extend our findings to another tumor model, we also tested the protective potential of previously-described CT26 colon carcinoma neo-epitopes[15] as we did for the B16F10Muts and found that 3 of the antigens tested (CT26Mut20, CT26Mut27, and CT26Mut37) could slow tumor growth in a therapeutic setting (Fig. 6e). When using Ad and MRB as adjuvants together with these 3 Muts to treat CT26 SC tumor-bearing animals, we observed that the animals mounted antigen-specific immunity against the 3 Muts (Fig. 6f), that their tumors grew slower (Supplementary Fig. 9) and that more than 20% of the vaccinated mice were cured (Fig. 6g). Taken together, our data show that OVs co-administered with tailored cocktails of peptidic cancer neo-antigens constitute efficient personalized anticancer vaccines.

Given that shared tumor antigen-encoding OVs are efficient treatments and that they could serve as adjuvant platforms for personalized vaccination, we tested the therapeutic use of OV-antigens co-administered with additional antigens (treatment schedule shown in Fig. 7a). We found that the Ova-specific immune response mounted following vaccination using Ad-Ova and MRB-Ova was similar to the one obtained using Ad-Ova +DCT and MRB-Ova+DCT (Fig. 7b). Furthermore, this later combination also allowed for the additional generation of DCT-specific immunity (dashed bars). We next tested whether the additional vaccination against Muts would confer superior

therapeutic benefits. To do so, we implanted SC B16F10-Ova tumors and immunized the animals with Ova-encoding Ad and MRB with or without co-injecting the 4 protective B16F10Muts. As previously published[16], our results show that the OV-Ova prime-boost could slow tumor growth and prolong survival, but all the animals still succumbed to the disease (Fig. 7c, left and right panels, respectively). On the other hand, the tumors from the group treated with OV-Ova+Muts grew slower and the survival was further improved, with 30% of the mice being cured. These results support the use of OVs encoding antigens personalized with additional cancer neo-epitopes as vaccination platforms.

## Discussion

Our strategy of co-administering antigenic peptides with OVs confers several advantages over the use of OVs encoding antigens. First, a previous report has shown that MRB encoding an antigen could efficiently boost immunity only when given IV[3], while our data demonstrates that MRB + antigen is as efficient IT as it is IV. This represents an important advantage of our strategy given that the direct injection of OVs in the tumors allows for the delivery of the entire viral dose to the cancer site, which could be a desired approach for some patients[17]. Second, since the antigen is administered as peptides, it is readily available for the immune system to recognize. Also, antigen expression is not driven by the

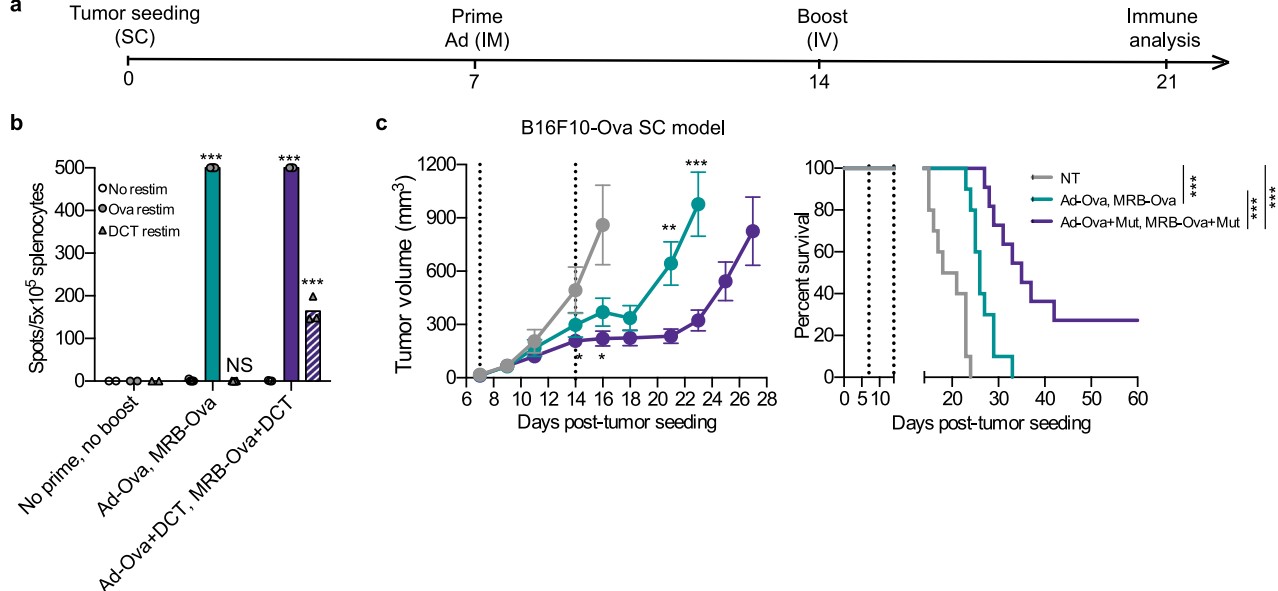

**Fig. 7 OVs encoding antigens can be used as platforms for personalized vaccines. a** Treatment schedule used in this study. **b** IFNγ ELISPOT analysis of splenocytes from mice primed with Ad-Ova ± DCT peptide and boosted with MRB-Ova ± DCT peptide (from left to right; $n = 2$, 5 and 3). The statistical analyses refer to the comparison between the corresponding ex-vivo "No restim" and "restim" conditions. NS: $p > 0.05$, ***: $p < 0.001$ (unpaired multiple two-tailed $t$-test). **c** Tumor growth and survival analyses of mice bearing established SC B16F10-Ova tumors treated with Ad-Ova and MRB-Ova together or not with B16Mut20, B16Mut30, B16Mut44 and B16Mut48 ($n = 10$). Data are presented as mean values ± SEM. *: $p < 0.05$, ***: $p < 0.001$ (unpaired multiple two-tailed $t$-test). Kaplan–Meier survival analysis: ***: $p < 0.001$ (Mantel–Cox test, two-sided). Source data are provided as a Source Data file. Exact $p$ values can be found in the Source Data.

virus and is therefore independent of the intrinsic viral sensitivity of each cancer. Third, given that we show that Ad, MRB, VSV and VV can all be used as vaccine adjuvants, it would be possible to design vaccination strategies that include multiple vaccine boosts using different OVs. This could maximize anti-tumor immunity, but also the direct tumor killing given that the different viruses all have oncolytic activities. This is especially important when considering the immune responses generated against the virus vectors themselves[18], which could limit the potential to boost multiple times with the same viral vectors. Since several OVs are currently being tested in different patient cohorts, the results from these trials will be informative as to what is the best virus to use for different indications. Therefore, not only can our OV vaccines be customized to the cancer's unique mutanome, but the adjuvant platform could also be tailored to each patient by selecting the best OV based on the type and location of their cancer and therefore maximize the chances of positive outcomes.

One way to further improve vaccination efficacy would be to use OVs engineered to express immune-stimulating transgenes that promote the generation of memory T cells as adjuvants. Our OV vaccines promoted the generation of EM (CD44hi, CD62lo) CD8 T cells, along with a much smaller population of CM (CD44hi, CD62hi) CD8 T cells. While the contribution of these memory T cell subsets to the observed anti-tumor immune response remains to be determined, designing strategies that promote the generation of less differentiated CM T cells which have superior anti-tumour activity[19] may further increase the effectiveness of OV vaccines. Several transgenes have been shown by us and others to improve the efficacy of different OVs in direct cancer treatment scenarios (interferon-gamma[20], interleukin-12[21]) as well as in a vaccination strategy using irradiated cancer cells infected with OVs (granulocyte macrophage-colony-stimulating factor[22] and interleukin-12[23]). The vaccine adjuvant activities of these viruses remain to be tested and additional OVs

could be engineered to express cytokines such as interleukins-2, -7 and -21, which have themselves been used as vaccination adjuvants[24]. Evaluating the type of T cells generated by these various cytokines and selecting cytokines that promote the generation of tumor-specific CM T cells remains a promising area for future development.

The use of OVs as vaccine adjuvants confers a rapid, easy and economical way to induce antigen-specific immunity using synthetic peptides. In this study, we found the OV MRB to be better at boosting antigen-specific immunity compared to other adjuvants used for vaccination in humans. Importantly, alum-based and oil-in-water adjuvants are usually used to induce humoral-rather than cell-mediated immunity and we did not assess the humoral immune response in our study. Notably, Addavax has been shown to also trigger antigen-specific T cells[25] and we did observe a modest but insignificant boosting effect using this adjuvant. Since the anti-Mut immunization must be directed against the mutated epitopes only in order to prevent the development of auto-immunity, the use of peptides is ideally suited for personalized vaccination. However, the use of OVs encoding tumor antigens is equally effective and might be preferable for the generation of multi-epitope immune responses against whole proteins such as shared cancer antigens. Therefore, our strategy is best suited for anticancer vaccines targeting the tumor mutanome. Importantly, our results show that both vaccination strategies are compatible. Therefore, OVs already encoding tumor antigens such as the clinical candidates Ad-MAGEA3 and MRB-MAGEA3 or Ad-E6/E7 and MRB-E6/E7 could easily be used as vaccination adjuvants to target additional Muts in MAGEA3- or E6/E7-positive tumors, respectively.

The heterologous OV prime-boost vaccination approach using viruses encoding tumor antigens is currently undergoing clinical investigation and thus the dosing and safety data will soon be established. Some challenges remain before considering translating OV-adjuvant vaccines to the clinic. Amongst those, we believe that

maximal benefits would be provided by anticancer vaccination if administered early in the course of the disease, whereas OV trials enroll patients with advanced cancers. Furthermore, anti-Mut vaccination is also being clinically investigated and therefore, we believe that the treatment strategy developed in this study could be rapidly translated to the clinic and could significantly improve the outcome of cancer patients. The OV-adjuvant platform is thus a feasible strategy to vaccinate patients against their unique cancer mutanome to induce powerful tumor-specific immune responses while directly killing tumor cells and warrants clinical testing.

## Methods

**Study design**. The objective of this study was to design an easily adaptable and efficient platform for anticancer vaccination. By using model antigens and cancer neo-epitopes, as well as various OVs such as Ad, MRB, VSV, VV and engineered viruses, we investigated the feasibility of priming and boosting anti-cancer immune responses using OVs co-administered with peptides and compared this approach to a prime-boost approach using OVs encoding the same antigens. All animals were included in the analysis.

**Cell lines and culture**. The CT26, B16F10, Vero, 293X, U2OS and HeLa cell lines were obtained from ATCC. All cell lines were maintained in Dulbecco's Modified Eagle's Medium (Corning Cellgro, 10013CV) supplemented with 10% fetal bovine serum (Sigma Life Science, F2442) and cultured at 37 °C with 5% $CO_2$.

**Virus production and quantification**. The MRB virus used in this study is the clinical candidate variant MG1[4]. The VSV used in this study is the attenuated oncolytic variant VSVΔ51 of the Indiana strain[26]. VSV and MRB were propagated, purified and quantified on Vero cells as described previously[20]. Briefly, virus stocks were purified from cell culture supernatants by filtration through a 0.22 μm Steritop filter (Millipore) and centrifugation at $30,000 \times g$ before resuspension in PBS. The E1/E3-deleted human type 5 Ads used in this study (Ad, Ad-Ova[16] and Ad-DCT[27]) are replication incompetent. For amplification of Ads, 80% confluent 293X cells were infected at an MOI of 1 for 72 h. Cell-associated virus was then collected by repeat[3] freeze-thaw cycles. The virus was then isolated by centrifugation for 3.5 h at $21,000 \times g$ through a CsCl gradient ($1.4–1.2$ g/cm³ CsCl in 10 mM Tris/HCl). The VV used in this study is the wild type Copenhagen strain and was produced in HeLa cells and quantified in U2OS cells[28]. For VV purification, cell-associated virus was collected by repeat[3] freeze-thaw cycles. Further purification of viral stocks was done by centrifugation at $20,700 \times g$ through a 36% sucrose cushion (in 1 mM Tris) before resuspension in 1 mM Tris, pH 9.

**Virus irradiation**. A Spectrolinker XL-1000 UV crosslinker was used to UV-inactivate MRB (2 min at 120 mJ/cm²) as done previously[29]. This irradiation results in the crosslinking of the viral genomes, which prevents replication.

**In vivo experiments and tumor models**. All experiments were approved by the University of Ottawa Animal Care Committee (ACC). SC tumors: $10^6$ B16F10, B16F10-Ova or CT26 cells were injected into the left flank of 6–8 week old female C57BL/6 or BALB/c mice, respectively (Charles River Laboratories). Lung tumors: $10^6$ B16F10 cells were injected in the tail vein of 8-week-old C57BL/6 mice. Unless specified otherwise, Ad ($1 \times 10^8$ plaque forming units (PFU)) was administered IM in the quadriceps and MRB, VSV and VV (all at a dose of $1 \times 10^8$ PFU) were administered IV via the tail vein. PolyI:C was purchased from Invivogen (31852-29-6) and used at a dose of 50 μg/animal/immunization. Addavax and Alhydrogel were both purchased from InvivoGen. Both adjuvants were mixed 1:1 with peptides and the Alhydrogel-peptide mix was incubated at room temperature for 1 h with rotation prior to vaccination. The peptides (100 μg/mouse/immunization) were pre-mixed with the different viruses or with polyI:C prior to injection in a total volume of 100 μL of PBS. Unless specified otherwise, immune priming and boosting were performed 7 and 14 days post-tumor seeding, respectively, and the immune analysis was performed 7 days after the last immunization. For CD8 depletion experiments, 200 μg of rat IgG2a, k anti-mouse CD8a (clone 53.67) or rat IgG2a isotype control antibodies (both from Leinco) were injected IP on days 7 and 14.

**Dot blot**. A $1 \times 10^8$ PFU of MRB and 100 μg of myc peptide mixture was filtered using 50 kDa cutoff Centricon filters as per the manufacturer's instructions. The retentate and the filtrate were collected for analysis. 5 μL drops of each were spotted onto Immun-blot 0.22 μm PVDF membranes (Bio-Rad) pre-activated with methanol for 5 min. Membranes were dried, re-hydrated with TBS-tween, blocked using a solution of 5% milk in TBS-tween and probed with anti-myc (Cell Signaling Technology, 1:1000) and anti-MRB (home-made, 1:3000)[29] antibodies. The peroxidase-coupled secondary antibodies were purchased from Molecular Probes (goat anti-rabbit and anti-mouse, both used at 1:1000). Between each step, the membranes were washed three times with TBS-tween without agitation. Signals were revealed using the Immobilon Forte western HRP substrate (Millipore).

**Immune analysis and peptides**. The treatment schedules are depicted in each figure. Flow cytometry: ex-vivo re-stimulations of splenocytes and flow cytometry staining were performed as described previously with the peptides indicated in the figures[16]. Briefly, single-cell suspensions were surface stained with fluorescently conjugated antibodies. Cell viability was assessed with Fixable Viability Dye eFluor 780 (1:1000) or eFluor 506 (1:750) according to manufacturer's protocols. For intracellular cytokine staining, splenocytes were stimulated ex-vivo with 1 μM OVA peptide for 5 h in the presence of GolgiStop (BD Biosciences). After restimulation, cells were surface stained, fixed and permeabilized using FoxP3/Transcription Factor Staining Buffer Set (BD Biosciences), stained for intracellular proteins, and analyzed by flow cytometry using an LSR Fortessa (BD Biosciences) cytometer. ELISPOT: mouse IFNγ ELISPOTs (MabTech, 3321-4AST-10) were performed according to the manufacturer's protocol. For flow cytometry analysis of blood, blood was collected from the tail vein and added into PBS supplemented with 2% FBS, 5 mM EDTA and 0.02% sodium azide. Red blood cells were lysed with ACK lysis buffer before staining. The antibodies used are: anti-mouse CD62L (clone MEL-14, 1:300), anti-mouse TNF-alpha (clone MP6-XT22, 1:250), anti-mouse IFN-gamma (clone XMG1.2, 1:250), anti-mouse CD8 (clone 53-6.7, 1:300), anti-mouse KLRG1 (clone 2F1, 1:300), anti-mouse CD44 (clone IM7, 1:300), anti-mouse CD21/CD35 (clone 7E9, 1:100), anti-mouse CD23 (clone B3B4, 1:300), anti-mouse B220 (RA3-6B2, 1:300) and anti-mouse CD19 (clone ID3, 1:300) were all purchased from eBioscience, anti-mouse CD127 (clone A7R34, 1:100) was purchased from BioLegend and H-2Kb/OVA257 tetramers were purchased from Baylor College of Medicine (1:100). The gating strategies are shown in Supplementary Fig. 10. The data were analyzed using FlowJo version 10.5.3.

All peptides were synthesized by Biomer Technology:
Ova: SIINFEKL
DCT: SVYDFFVWL
Myc: EQKLISEEDL
B16Mut05: FVVKAYLPVNESFAFTADLRSNTGGQA
B16Mut17: VVDRNPQFLDPVLAYLMKGLCEKPLAS
B16Mut20: FRRKAFLHWYTGEAMDEMEFTEAESNM
B16Mut22: PKPDFSQLQRNILPSNPRVTRFHINWD
B16Mut25: STANYNTSHLNNDVWQIFENPVDWKEK
B16Mut28: NIEGIDKLTQLKKPFLVNNKINKIENI
B16Mut30: PSKPSFQEFVDWENVSPELNSTDQPFL
B16Mut44: EFKHIKAFDRTFANNPGPMVVFATPGM
B16Mut48: SHCHWNDLAVIPAGVVHNWDFEPRKVS
CT26Mut03: DKPLRRNNSYTSYIMAICGMPLDSFRA
CT26Mut20: PLLPFYPPDEALEIGLELNSSALPPTE
CT26Mut26: VILPQAPSGPSYATYLQPAQAQMLTPP
CT26Mut27: EHIHRAGGLFVADAIQVGFGRIGKHFW
CT26Mut37: EVIQTSKYYMRDVIAIESAWLLELAPH

**Statistical analysis**. The statistical analyses were performed using GraphPad Prism 6.0e. as depicted in the figure legends. Error bars represent the standard error of the mean.

**Statistics and reproducibility**. All findings were reproducible. Experiments were independently performed the following number of times: Fig. 1b (2), Fig. 1c (2), 1d (2), 1e (2), 1f (2), 1g (2), 2b (2), 2c (4), 2d (3), 2e (1), 2f (1, 2 experiments combined), 3b (2), 3c (2), 3d (2), 3e (2), 3f (2), 3g (2), 3h (2), 3i (2), 4b (4), 4c (2), 3d (1), 4e (1, 2 experiments combined), 5b (1), 5c (1), 5d (1), 5e (1), 5f (2), 6b (1, 2 experiments combined), 6c (1), 6d (2), 6e (1, 2 experiments combined), 6f (1), 6g (1), 7b (2), 7c (2), Supplementary Fig. 1 (2), 2a (2), 2b (2), 3 (1), 4 (1), 5 (1), 6 (1), 7 (2), 8a (1), 8b (1), 9 (2).

**Reporting summary**. Further information on research design is available in the Nature Research Reporting Summary linked to this article.

## Data availability

Source data are provided with this paper. All data generated during and/or analyzed during this study are available from the corresponding author on request. Source data are provided with this paper.

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

## Acknowledgements

Supported by: the Canadian Institute for Health Research, the Terry Fox Foundation, the Canadian Cancer Society Research Institute, the Ontario Institute for Cancer Research, the Ottawa Regional Cancer Foundation, the Ottawa Hospital Foundation, BioCanRx, the Alliance for Cancer Gene Therapy, the "Institut du cancer de Montréal", the "Fonds de Recherche Québec-Santé" and the "Fondation Québécoise du Cancer du Sein".

## Author contributions

M.C.B.D designed the study, performed experiments and wrote the paper. D.G.R., S.T.K., K.G., N.T.M., M.M., A.S.A., J.K, D.B., K.S. and C.T.d.S. performed experiments. The paper was written and revised by M.C.B.D., D.G.R., R.C.A., B.D.L., D.F.S. and J.C.B.

## Competing interests

J.C.B., B.D.L. and D.F.S. have equity interest in Turnstone Biologics, which develops MRB as an OV platform and M.C.B.D. and J.C.B. are inventors on a patent describing the use of OVs as adjuvants for anticancer vaccination. All other authors declare no competing interests. All data associated with this study are available in the main text or the Supplementary Information.
