## [Peer Review File · Nature Communications]

Reviewers' comments:

Reviewer #1 (Remarks to the Author); expert in oncolytic virus and mouse models:

The present study by Roy et al is an innovative work and provides surprising and important information on the combination of virotherapy and vaccinations with strong implications for future design of clinical vaccination trials. The study first shows that OV's are effective adjuvants for peptide vaccinations during prime and in heterologous prime-boost schemes. These treatments lead to improved survival in a B16 model of established lung colonies. An interesting finding was that replication was not required for this adjuvant property. The authors furthermore showed that OV and vaccine combination was able to effectively elicit T cells against neoepitopes (and panels thereof) yielding a therapeutic benefit by inhibiting tumor growth and improving survival. In a final approach the authors demonstrated that OV's expressing shared antigens could be used as adjuvant platform for neoepitope vaccines to address both pan-tumor and personalized targets.

A central message of the paper is that the use of a prime-boost regimen using heterologous oncolytic viruses combined with tumor-antigen directed vaccine seem to work as efficient like the use of equivalent viruses that express the tumor antigen. The presented method is particularly promising with regard to vaccines targeting cancer-specific neoepitopes or neoepitope panels. Mutation-derived neoepitopes are more or less unique in patients and the need for patient-specific vectorization is an enormous hurdle for broad clinical application. The present study by Roy et al offers an elegant and simple solution. The overall technical quality of the paper is excellent. I have only a few concerns:

1. The authors show in Fig 4 B that PolyI:C and MRB are comparable in inducing Mut30-specific T cell responses (Fig. 4B). Furthermore the authors show in Fig. 3B that the Maraba alone group (only oncolysis) was ineffective in the B16 lung model. In light of these observations it is astonishing that the effect of Maraba with Mut30 vaccine in Fig. 4E is dramatically better compared with polyI:C and Mut30. How is the efficacy of Maraba alone in that model?

2. A more detailed technical description (or reference) of the used viruses is required. Exemplarily, the technical setup of the used Ad-DCT and the Ad control and the replication capability of these viruses is unclear. At least in the cited references I can only find the description of an Ad-DCT that is E1/E3 deleted and thus replication-incompetent. Should be clarified and modified if necessary.

3. The numbers of used animals per group should be provided in the figure legends.

Suggestions:

4. Though it is mentioned in the methods section it would make comparison of growth/survival curves more easier when the start of therapy would be annotated in all figures.

5. Citation 2 is a follow up paper on the VSV prime/Ad boost scheme by the Wan group. The initial description would also fit (Bridle et al., Mol Ther, 2009). There is some space in general for more literature and more mechanistic discussion.

Reviewer #2 (Remarks to the Author); expert in tumour antigens, immunotherapy and oncolytic virus:

This manuscript builds on previous data from this group showing that a prime boost strategy using two different viruses (Ad and Maraba) can lead to the generation of potent anti tumor T cell responses against an antigen encoded by the viruses. Here, the authors extend this by showing that the antigen can be vaccinated against by providing it as a peptide in conjunction with the virus, without actually having to have it encoded by the virus. This is potentially an important technological advance because it means that the virus can be used as an adjuvant with multiple peptides, including peptides representing neo-antigens – thereby obviating the need to make viruses with multiple transgenes or multiple viruses.

Specific Comments:

Fig.1: Why is MRB+DCT a better priming agent than MRB-DCT? There is no mechanism even attempted for this effect.

Fig.1C: It is surprising that the anti-OVA response is about the same as the anti-DCT response in these mice since one is a clearly foreign antigen and one is a self antigen. Can the authors explain this?

Fig.2: The prime boost with MRB-DCT or MRB+DCT lead to apparently very similar levels of T cell reactivity which are presumably CD8+ T cells as this is an ELISPOT assay. The only really significant mechanism provided throughout the manuscript is ELISPOT readouts.

However, it is important to show the mechanisms of in vivo action and effect of the virus+peptide and simply providing ELISPOT assays does not provide that. Are the in vivo effectors CD8, CD4 or other types of cells? Are the same cells activated by the virus encoded boost compared to the peptide + virus?

Where does the virus go either with encoded antigen or with added peptide? Are these the same pathways of antigen presentation and T cell re-stimulation?

Does the virus go to the tumor? Is there any contribution of oncolysis. There are no mechanisms presented beyond these ELISPOTs and that significantly reduces the impact.

Fig.3: Where are the controls of Ad-DCT + MRB-DCT which is the gold standard of what this group have published before? These controls are very important if the claim of at least equivalence of peptide + virus compared to virus-encoded antigen is to be supported.

Figs.4: The in vivo data lack controls and lacks any mechanism of how the tumor control is mediated in vivo.

Fig.5: The in vivo data lack controls and lacks any mechanism of how the tumor control is mediated in vivo. What happens if the mice are treated with the mut peptide mix alone or with a different adjuvant? These controls are very important.

Fig.6: Ad-ova+MRB-ova leads to some therapy which is enhanced by adding the cocktail of B16 Mut peptides. There should be control peptides added to the virus mix as well. What are the mechanisms of tumor control? CD8, CD4, NK?

The concept of prime boost against antigens is well established. This group has established the same immunological principle with the Ad-Maraba combination. Therefore, the

immunological advance reported here is relatively small. The fact that it is possible to boost a pre-primed response with a peptide+adjuvant instead of a virus encoded antigen is not itself very surprising. As such the advance is more technical than conceptual. In addition, the authors present minimal mechanistic data to explain the role of the virus as the adjuvant. There is no information on how the Maraba virus mediates immunological boosting – is the peptide bound to the virus? Does the virus travel to LN or is it activating the critical APC previously described to be important for Maraba-Ag presentation? Does the iv administered virus go to the tumor and achieve any oncolysis or are the immune effects described all exclusively systemic. In addition, the authors do not present any data to indicate that a boost of peptide+virus is the same as, better than, or worse than using other adjuvants that are already approved for use in humans. This is an important point because if the prime boost strategy is to be used clinically (as it already is with virus encoded tumor antigens) there should be a good rationale for the need for virus as the adjuvant over other possible adjuvants.

Reviewer #1 (Remarks to the Author); expert in oncolytic virus and mouse models:

“The present study by Roy et al is an innovative work and provides surprising and important information on the combination of virotherapy and vaccinations with strong implications for future design of clinical vaccination trials. The study first shows that OVs are effective adjuvants for peptide vaccinations during prime and in heterologous prime-boost schemes. These treatments lead to improved survival in a B16 model of established lung colonies. An interesting finding was that replication was not required for this adjuvant property. The authors furthermore showed that OV and vaccine combination was able to effectively elicit T cells against neoepitopes (and panels thereof) yielding a therapeutic benefit by inhibiting tumor growth and improving survival. In a final approach the authors demonstrated that OVs expressing shared antigens could be used as adjuvant platform for neoepitope vaccines to address both pan-tumor and personalized targets.

A central message of the paper is that the use of a prime-boost regimen using heterologous oncolytic viruses combined with tumor-antigen directed vaccine seem to work as efficient like the use of equivalent viruses that express the tumor antigen. The presented method is particularly promising with regard to vaccines targeting cancer-specific neoepitopes or neoepitope panels. Mutation-derived neoepitopes are more or less unique in patients and the need for patient-specific vectorization is an enormous hurdle for broad clinical application. The present study by Roy et al offers an elegant and simple solution. The overall technical quality of the paper is excellent. I have only a few concerns:”

1.1. *“The authors show in Fig 4 B that PolyI:C and MRB are comparable in inducing Mut30-specific T cell responses (Fig. 4B-now 5C). Furthermore the authors show in Fig. 3B (now 4B) that the Maraba alone group (only oncolysis) was ineffective in the B16 lung model. In light of these observations it is astonishing that the effect of Maraba with Mut30 vaccine in Fig. 4E (now 5F) is dramatically better compared with polyI:C and Mut30. How is the efficacy of Maraba alone in that model?”*

- Please note that the figure labels have changed. Here, we will refer to the new figure numbers. The reviewer raises a valid point, Figure 5C does show that polyI:C and MRB induce comparable anti-Mut30 immune responses and figure 5F shows that the effect of the MRB vaccine is dramatically better than that of polyI:C while figure 4B shows that MRB alone is ineffective (similar results were also obtained in figure 6D). However, we believe that it is difficult to directly compare the results obtained from these experiments as the tumor models and treatment schedules used are different. For instance, the lung model used in figure 4B is extremely aggressive and the animals received MRB treatment 14 days after tumor seeding, a time at which lung tumors are well established and very difficult to treat. In that experiment, MRB alone did not lead to any prolongation of survival. The subcutaneous B16F10 tumor model used in figure 5F is also very aggressive, but the animals were treated 7 days earlier, a time at which all tumors are very small and therefore easier to control (the tumor growth curves on the left panel shows the first measures at day 10). Although we did not investigate the contribution of direct oncolysis, as it would be difficult because the tumors are extremely small at that time point with most of them having an average diameter of 1mm, we believe that it plays an important role in the tumor control of the animals vaccinated with MRB in figure 5F.

1.2. *“A more detailed technical description (or reference) of the used viruses is required. Exemplarily, the technical setup of the used Ad-DCT and the Ad control and the replication capability of these viruses is unclear. At least in the cited references I can only find the description of an Ad-DCT that is E1/E3 deleted and thus replication-incompetent. Should be clarified and modified if necessary.”*

- The reviewer is correct, the Ad we used in our study is E1/E3 deleted. This is extremely important and we have now added more information in the methods section of the manuscript. We would like to thank the reviewer for pointing this out.

1.3. *“The numbers of used animals per group should be provided in the figure legends.”*
- We have now added the number of mice used in each experiment to the figure legends. We often used only 2 animals as controls for the immune analysis because we consistently obtain no background.

Suggestions:

1.4. *“Though it is mentioned in the methods section it would make comparison of growth/survival curves more easier when the start of therapy would be annotated in all figures.”*

- Given that the experimental design changes from figure to figure, we understand that it can be confusing for the reader. To facilitate the interpretation of the data, we have now included schematic representations of the treatment schedules used in each figures (new figures 3A, 4A, 5A, 6A and 7A). The figure legends and methods section also include the information. For tumor growth and survival graph, the days of treatment are also indicated by dashed lines on each graph. We believe that it is now easier to compare the results obtained from different experiments.

1.5. *“Citation 2 is a follow up paper on the VSV prime/Ad boost scheme by the Wan group. The initial description would also fit (Bridle et al., Mol Ther, 2009). There is some space in general for more literature and more mechanistic discussion.”*

- The reviewer is right; we were indeed referring to a follow up paper about the oncolytic virus prime-boost strategy instead of the initial study by Bridle et. al published in 2009. We initially chose to cite the follow up study because, as we do in our study, it uses Ad as a priming agent instead of a boost as in the original 2009 paper. We acknowledge that both studies are important and now refer both papers in our manuscript.

In regards to mechanistic discussions, the revised version of our manuscript now includes a longer discussion about the mechanisms at play. As explained in more details below in our response to Reviewer 2, we now discuss additional aspects of our vaccines such as the biodistribution of MRB with and without peptide (2.4), the infection of splenic B cells (2.11), the generation of different subsets of memory T cells (2.3) and a cell depletion experiment (2.7). These are aspects that we addressed experimentally and the results obtained allowed us to understand and discuss the mechanisms at play by our vaccine.

Reviewer #2 (Remarks to the Author); expert in tumour antigens, immunotherapy and oncolytic virus:

“This manuscript builds on previous data from this group showing that a prime boost strategy using two different viruses (Ad and Maraba) can lead to the generation of potent anti tumor T cell responses against an antigen encoded by the viruses. Here, the authors extend this by showing that the antigen can be vaccinated against by providing it as a peptide in conjunction with the virus, without actually having to have it encoded by the virus. This is potentially an important technological advance because it means that the virus can be used as an adjuvant with multiple peptides, including peptides representing neo-antigens – thereby obviating the need to make viruses with multiple transgenes or multiple viruses.”

Specific Comments:

2.1. *“Fig.1: Why is MRB+DCT a better priming agent than MRB-DCT? There is no mechanism even attempted for this effect.”*

- This is an interesting question. While the inability of MRB-antigen to prime an antigen-specific immune response has already been described by the group of Lichty in 2014 (a study we refer to in our manuscript), we found in our study that the co-administration of MRB with peptides does induce

antigen-specific immunity. For MRB-antigen, the antigen is encoded by the viral genome. The expression of the target antigen thus requires proficient infection. When treating mice early (such as 7 days post-tumor seeding like we do in our experiment), there might not be sufficient replication and therefore enough antigen production to induce an immune response directed against the encoded antigen. This is a situation that might also be observed when using virus-resistant tumor models. In sharp contrast, our approach of co-administering the virus with peptide ensures that sufficient amounts of antigens will be present regardless of the tumor burden or the intrinsic viral susceptibility of each cancer. We believe that the difference resides in the rapid and sufficient availability of antigen using our approach and now discuss this aspect in the discussion section of our manuscript.

2.2. *“Fig.1C: It is surprising that the anti-OVA response is about the same as the anti-DCT response in these mice since one is a clearly foreign antigen and one is a self antigen. Can the authors explain this?”*

- We agree with the reviewer that the immune response directed against the foreign antigen Ova should be greater than the one against DCT. The ELISPOT results from the DCT and Ova models shown in Fig. 1C were obtained from different experiments. It is therefore difficult to compare the magnitude of the responses. Nevertheless, we did observe differences. Although we understand that the responses might seem equivalent, the signal obtained for Ova was twice as strong compared to the anti-DCT vaccination (means of 124 vs 63 spots, respectively). Furthermore, when comparing the anti-DCT to anti-Ova immune responses in prime-boost vaccination regimens, we also observed greater anti-Ova than anti-DCT responses (Figs. 2C vs D and 4D left and right panels).

We therefore observe greater immunity against Ova compared to DCT, which was expected and supports the comment of the reviewer.

2.3. *“Fig.2: The prime boost with MRB-DCT or MRB+DCT lead to apparently very similar levels of T cell reactivity which are presumably CD8+ T cells as this is an ELISPOT assay. The only really significant mechanism provided throughout the manuscript is ELISPOT readouts. However, it is important to show the mechanisms of in vivo action and effect of the virus+peptide and simply providing ELISPOT assays does not provide that. Are the in vivo effectors CD8, CD4 or other types of cells? Are the same cells activated by the virus encoded boost compared to the peptide + virus?”*

- The reviewer is raising an important point. ELISPOT readouts are indeed not sufficient to characterize the immune cells induced by our vaccine. Our original submission did include flow cytometry (Figs. 2E and F) showing that CD8 T cells from vaccinated animals produce IFN γ upon *ex-vivo* re-stimulation with peptide. Although this information is important, we agree with the reviewer that a deeper characterization of these cells would benefit our study. We therefore performed additional experiments in which we characterized antigen-specific CD8 T cells (Fig. 3). We analyzed blood samples collected at 3 different time points (14, 28 and 35 days post-prime) as well as splenocytes at day 35. Using tetramers, we found an increased proportion of antigen-specific CD8 T cells in the blood of animals that were boosted using MRB+antigen and the cell numbers remained high over time (more than 50% of blood CD8 T cells were antigen specific even at day 35). Cytokine production from splenocytes re-stimulated *ex-vivo* at day 35 confirmed the results obtained in Fig. 2E showing that vaccinated animals have CD8 T cells that react to the antigen and further revealed that even 4 weeks after boost, 10-15% of splenic CD8 T cells have the ability to produce cytokines upon re-stimulation with antigen. Based on KLRG1 and CD127 expression, we found that the main populations of antigen-specific cells generated by our vaccine were short-lived effector cells (SLECs, KLRG1^{hi}, CD127^{lo}), double-positive effector cells (DPECs, KLRG1^{hi}, CD127^{hi}) and memory precursor effector cells (MPECs, KLRG1^{lo}, CD127^{hi}). We also observed effector memory and central memory cells to be generated to important amounts with vaccination. These new data are included in Fig. 3 and discussed in the manuscript.

Importantly, we also performed an experiment in which animals were vaccinated and CD8 T cells were depleted (Fig. 4E) and found that CD8 T cells were required for the therapeutic efficacy of our

vaccine. This experiment is described in further details below (2.7).

2.4. *“Where does the virus go either with encoded antigen or with added peptide? Are these the same pathways of antigen presentation and T cell re-stimulation?”*

- The reviewer raises another important point. We have no reason to believe that the biodistribution of the virus would be different with or without peptide co-administration. To answer this question, we performed an experiment in which we intravenously injected MRB-Ova either alone or together with Ova peptides. We used MRB encoding Ova for both groups in order to avoid any error due to differences in the amount of virus administered. We collected different organs 24h after injection. We chose this time point because we know from previous work that we can still detect MRB from these organs at 24h. Our results show that the pre-mixing of peptides with the virus does not modify its biodistribution. This experiment is now shown in Fig. 1E.

2.5. *“Does the virus go to the tumor? Is there any contribution of oncolysis. There are no mechanisms presented beyond these ELISPOTs and that significantly reduces the impact.”*

- As explained above (2.4), we performed a biodistribution experiment (Fig. 1E) and did not observe any differences in the amount of virus found in normal organs with the pre-mixing of peptides. We therefore have no reason to believe that the delivery of MRB to the tumors would be prevented by the addition of peptide.

As shown by our biodistribution experiment, MRB is found in the lungs 24h post-injection and that is independent of the presence of lung tumors. We did not directly assess oncolysis post-MRB treatment in our vaccination model because it would be difficult to discriminate between immune-mediated and virus-mediated tumor killing given that the mice are already primed against a tumor antigen. However, previous work has shown that MRB titers were 100-1000 times higher in the lungs of B16F10 lung tumor-bearing animals 11-48h post-injection compared to tumor-free mice (Pol et. al., 2014; PMID: 24322333, Fig. 1D), therefore indicating that MRB replicates in established lung tumors. This is particularly relevant as it uses the B16F10 lung tumor model and the animals were treated with MRB 14 days post-tumor seeding, as we do for immune boosting in our study.

2.6. *“Fig.3 (now Fig 4): Where are the controls of Ad-DCT + MRB-DCT which is the gold standard of what this group have published before? These controls are very important if the claim of at least equivalence of peptide + virus compared to virus-encoded antigen is to be supported.”*

- We agree with the reviewer that this is an important control. We therefore performed the experiment and, as expected, found no difference in therapeutic efficacy when comparing both vaccination approaches (Fig. 4D). This new data further strengthens our initial claim that both approaches are equivalent. We would like to thank the reviewer for the suggestion.

2.7. *“Figs.4 (now Fig. 5): The in vivo data lack controls and lacks any mechanism of how the tumor control is mediated in vivo.”*

- Our original manuscript did lack mechanistic studies. In addition to the new immune characterization described in 2.3 (Fig. 3), we also performed a cell depletion experiment to identify the mediators of tumor control post-vaccination. Given that we identified CD8 T cells as being massively induced by our vaccine (Figs. 2E, F and 3), we performed a survival study using the B16F10 lung tumor model in which CD8 T cells were either depleted or not (Fig. 4E). Our results show that the protection conferred by Ad+Ova and MRB+Ova adjuvant vaccination was almost completely lost when CD8 T cells are depleted. This experiment confirms a crucial role for CD8 T cells in the therapeutic efficacy of adjuvant oncolytic vaccination, which is now discussed in the paper.

2.8. *“Fig.5 (now Fig. 6): The in vivo data lack controls and lacks any mechanism of how the tumor*

control is mediated in vivo. What happens if the mice are treated with the mut peptide mix alone or with a different adjuvant? These controls are very important.”

- Although we did not investigate this aspect with Mut peptide, we did test if the administration of Ova peptide alone was sufficient at boosting anti-Ova immunity primed by Ad-Ova (Fig. 1D and E). The ELISPOT and flow cytometry results we obtained both show that the Ova peptide alone fails at boosting antigen-specific immunity.

We agree with the reviewer that comparing the adjuvant activity of the virus with the ones of other adjuvants is an important point. Our initial study only included data comparing the adjuvant activity of MRB with that of polyI:C (Figs. 5C) and we found no difference. This was done with Mut peptides. Although polyI:C is a commonly used adjuvant in pre-clinical vaccination studies, early trials found that its toxicity was too high to be used in cancer patients (Krown et. al, 1985. PMID: 2418162). In order to compare MRB with adjuvants that could be used in the clinic, we tested its adjuvant activity head to head with Addavax, an oil-in-water emulsion, and Alhydrogel, an aluminum hydroxide-based adjuvant. Addavax, the equivalent of MF59, is an adjuvant that is used for the influenza vaccine and Alhydrogel is used in vaccines against many pathogens such as malaria. Our results show that only MRB can efficiently boost Ad-primed CD8 T cell immunity (Fig. S6). Importantly, and as we mention in the discussion section of the manuscript, while Addavax has been described as inducing T cell immunity, both adjuvants are usually used in vaccines to trigger antibody responses, an aspect that we did not investigate.

Taken together, our results show that MRB is better than polyI:C, Addavax and Alhydrogel at boosting antigen-specific T cell-mediated immunity in our heterologous vaccination regimen.

2.9. “Fig.6 (now Fig.7): Ad-ova+MRB-ova leads to some therapy which is enhanced by adding the cocktail of B16 Mut peptides. There should be control peptides added to the virus mix as well. What are the mechanisms of tumor control? CD8, CD4, NK?”

- As explained in 2.7, we have now performed an experiment in which CD8 T cells are depleted which leads to a complete loss of therapeutic efficacy. Although we did not perform this cell depletion experiment with Mut peptides, we have no reason to believe that the mechanism would be any different. Also, as explained in 2.8, the immunization of mice with peptides alone did not result in any immune boosting and should therefore not provide any therapeutic benefits. Furthermore, the immunization of subcutaneous B16F10 tumor-bearing mice with polyI:C+Mut30 failed at prolonging the survival of mice compared to polyI:C alone. We therefore believe that CD8 T cells are the mediators of our vaccine’s efficacy and that peptides alones would not be sufficient to confer therapeutic benefits.

2.10. “The concept of prime boost against antigens is well established. This group has established the same immunological principle with the Ad-Maraba combination. Therefore, the immunological advance reported here is relatively small. The fact that it is possible to boost a pre-primed response with a peptide+adjuvant instead of a virus encoded antigen is not itself very surprising. As such the advance is more technical than conceptual. In addition, the authors present minimal mechanistic data to explain the role of the virus as the adjuvant. There is no information on how the Maraba virus mediates immunological boosting – is the peptide bound to the virus?”

- We once again agree with the reviewer that mechanistic data was missing from our initial submission.

Regarding the binding of the peptide to the virus, we have performed an experiment in which we mixed virus with peptide as we do when we prepare our vaccine and then used centrifugal filters with a cutoff that is smaller than the virus, but bigger than the peptide. Therefore, free peptides would be found in the filtrate while the virus, together with any bound peptides, would be found in the retentate. The analysis of small proteins or peptides by western blot comprises many technical difficulties (migration, transfer through the membrane, weak binding to membranes that leads to detachment during washes). We therefore analyzed our samples by dot blot. Our results show that while MRB was

retained by the filter, we did not detect any peptide in the retentate along with MRB, and that the peptide was exclusively found in the filtrate, therefore indicating that it was not bound to the virus (Fig. S1).

We hope that the reviewer will agree that the new additions to our manuscript: the peptide binding described above, but also flow cytometry characterization (Fig. 3, discussed in 2.3), cell depletion (Fig. 4E, discussed in 2.7), biodistribution (Fig. 1E, discussed in 2.4), as well as the experiment described below (2.11) in which we assess the infection of splenic B cells (Fig. 1F and G) greatly complement our study and now provide sufficient conceptual and mechanistic advances.

2.11. *“Does the virus travel to LN or is it activating the critical APC previously described to be important for Maraba-Ag presentation?”*

- The reviewer is raising a very interesting question. We believe he is referring to a paper published in 2016 by the group of Wan (Bridle et. al., PMID: 27183620). In this study, the authors show that the infection of splenic B cells, more specifically follicular B cells, by an oncolytic virus encoding an antigen is important for the induction of antigen-specific immunity. Although the virus used in that study was VSV and not MRB, the 2 viruses are very closely related. We are not aware of another study in which the same was demonstrated for MRB. In order to determine if MRB would also infect these cells and if the presence of peptides would impair this infection, we performed an experiment that was similar to the one found in the 2016 paper. Briefly, MRB-GFP (with or without peptides) was administered to mice intravenously and the splenocytes were collected 1.5h and cultured ex-vivo for 4.5h to allow sufficient time for virus-driven GFP expression. As was observed in the 2016 study, MRB did infect splenic B cells, most of which were follicular B cells. Also, the presence of peptide did not impair this ability. This data can now be found in figures 1F and G.

2.12. *“Does the iv administered virus go to the tumor and achieve any oncolysis or are the immune effects described all exclusively systemic.”*

- As this point has previously been addressed, please refer to the discussion in section 2.5.

2.13. *“In addition, the authors do not present any data to indicate that a boost of peptide+virus is the same as, better than, or worse than using other adjuvants that are already approved for use in humans. This is an important point because if the prime boost strategy is to be used clinically (as it already is with virus encoded tumor antigens) there should be a good rationale for the need for virus as the adjuvant over other possible adjuvants.”*

- The reviewer is right, this is a critical point to address. We therefore performed an experiment in which we compared the adjuvant immune boosting activities of MRB, Addavax (an oil-in-water emulsion) and Alhydrogel (an aluminum hydroxide-based adjuvant) (as discussed in 2.8). We found that, not only was MRB the better adjuvant, but both Addavax and Alhydrogel failed at boosting antigen-specific CD8 T cells (Fig. S6). As mentioned above, as well as in the discussion of our manuscript, while Addavax has been described as inducing T cell immunity, both adjuvants are usually used in vaccines that trigger antibody responses, an aspect that we did not investigate.

We believe that the superior immune boosting capacity of MRB upon Ad priming, together with its cancer-killing ability justify its use as a vaccine adjuvant.

Reviewers' comments:

Reviewer #1 (Remarks to the Author):

my concerns have been appropriately addressed

Reviewer #2 (Remarks to the Author):

The authors have been highly responsive to my comments and requests. Thank you for taking these into account. i would support publication of this very interesting work.